# Achieving Precise Cup Positioning in Direct Anterior Total Hip Arthroplasty: A Narrative Review

**DOI:** 10.3390/medicina59020271

**Published:** 2023-01-31

**Authors:** Laura Elisa Streck, Friedrich Boettner

**Affiliations:** Hospital for Special Surgery, 535 East 70th Street, New York, NY 10021, USA

**Keywords:** hip replacement, safe zone, component positioning, fluoroscopy, navigation, imaging

## Abstract

Malpositioned implants in total hip arthroplasty are associated with impingement, increased wear, and dislocations, thus precise cup positioning is crucial. However, significant deviations between targeted and actually achieved cup positions have been found even in patients operated by experienced surgeons. When aiming for higher accuracy, various methods based on freehand positioning lead by anatomic landmarks, C-arm fluoroscopy, imageless navigation, or robotic-assisted-surgery have been described. There is a constant development of new products aiming to simplify and improve intraoperative guidance. Both the literature and expert opinions on this topic are often quite controversial. This article aims to give an overview of the different methods and systems with their specific advantages and potential pitfalls while also taking a look into the future of cup positioning in anterior hip replacements.

## 1. Introduction

Total hip arthroplasty (THA) has been called the most successful surgery of the century, with reported patient satisfaction rates of over 90% [1,2]. To achieve good results, proper implant positioning is essential. Malpositioning of the cup is associated with dislocation, psoas impingement, bony impingement limiting the range of motion, leg length discrepancy, and increased wear, problems that ultimately necessitate revision surgery [3,4,5,6].

The main variables used to describe the position of the cup are version and inclination (abduction). Version means the orientation of the cup in the sagittal plane. Higher anteversion allows for more flexion and adduction of the hip before impingement occurs. At the same time, higher anteversion may increase the risk for anterior dislocation while reduced anteversion or retroversion correlate with an increased risk for posterior dislocation. Inclination means the orientation of the cup in the coronal plane. High inclination (at maximum a vertical cup) leads to a smaller contact area at the superior dome of the cup and may increase contact stress possibly leading to increased implant wear. Low inclination on the contrary leads to earlier impingement, especially during abduction and therefore limits the range of motion.

Historically, placement of the cup has been guided according to the «safe zone» defined by Lewinnek et al. in 1978 [7]. The proposed zone was 30–50° inclination and 5–25° anteversion, aiming to minimize the risk for dislocation [7]. There is conflicting evidence on whether placement outside of this range is indeed related to higher dislocation rates [8,9,10,11,12]. Over the years, multiple slight variations of the safe zones have been described [13]. Weber et al. criticized that most methods could not prevent bony impingement and suggested smaller safe zones with best results between 40–50° inclination and 20–30° anteversion [6,14]. Other authors propagated concepts of a combined anteversion, including the positioning of the stem [15], or patient-specific target zones [16,17].

Independent of this discussion regarding the exact values and individual safe zones, implanting the cup in the desired position can be challenging. Although the target zone is reached in the majority of cases [18], even experienced surgeons report significant outliers [19,20,21]. The risk for malpositioned components may increase when performing surgery on the surgeon’s non-dominant side, when surgery is performed by low-volume surgeons, in patients with altered bony anatomy, or if patients have an increased body mass index (BMI) [21].

The extent to which the surgical approach effects the accuracy of cup positioning is under discussion. Some studies showed superior results for THA implanted via a direct anterior approach compared to posterior or lateral approaches and reported lower dislocation rates [22,23,24,25]. On the contrary, other authors reported comparable results, independent of the approach [26,27]. However, an advantage of the direct anterior approach is the supine positioning of the patient on the operating table which facilitates the utilization of intraoperative fluoroscopy.

This article aims to provide an overview of current concepts and methods for accurate cup positioning in primary anterior approach THA.

## 2. Cup Positioning Techniques

The organizational diagram in Figure 1 provides an overview of various cup positioning techniques in total hip arthroplasty.

### 2.1. Freehand Positioning Based on Anatomic Landmarks

Acetabular component placement can be based on anatomic landmarks. Various reference points are described, such as the anterior pelvic plane (formed by the anterior superior iliac spines and the pubic tubercules), the anterior and transverse acetabular notch or the transverse acetabular ligament [28,29,30,31]. It has been suggested that orienting the inferomedial rim of the cup parallel to the transverse acetabular ligament allows positioning in anteversion within the safe zones without using additional guidance [32,33]. With this technique, Archbold et al. reported a low dislocation of rate of 0.6% using a posterolateral approach [31]. Molho et al. found 100% of the hips within the safe zone for inclination and 90% within the safe zone for anteversion when referring to the transverse acetabular ligament [29]. However, Ling et al. found that pelvises with sagittal backward malrotation (roll back) were more prone to have cups implanted in increased inclination outside the safe zone [34]. The later also exemplifies that a perfect position based on anatomic landmarks can result in position outside the safe zone when measured on postoperative radiographs since roll back of the pelvis will increase inclination on standard radiographs.

In addition, even a minor failure to identify landmarks correctly can lead to inaccuracies in the final cup positioning [35]. Soft tissue contractures, anatomic changes due to osteophytes or previous trauma, or anatomic variances associated with hip dysplasia can cause errors in the correct identification of anatomic landmarks [32].

### 2.2. Positioning with Intraoperative C-Arm Imaging

Intraoperative fluoroscopy utilizing a C-arm is widely available. The supine positioning of the patient during anterior approach THA facilitates the intraoperative use of the C-arm to control component position. Measurement of angles can be altered by stereometric effects (parallax), which may lead to incongruency between intraoperative fluoroscopic and postoperative radiographic measurements [36,37].

#### 2.2.1. C-Arm Imaging without Additional Devices

Boettner and Rueckl found intraoperative fluoroscopy reliable to measure inclination with reference to the inter-teardrop line, when pelvic tilt is considered using the method described by Tannast et al. [38,39]. To adjust for parallax when measuring inclination, the authors recommended adding a correction factor of 5° to the intraoperative C-arm inclination measurement [38]. This number may vary depending on the C-arm model.

To obtain cup anteversion, various methods have been developed based on AP pelvic radiographs [7,40,41,42,43]. Methods described by Lewinnek [7], Hassan [41] and Liaw [42] have been proven to be reliable [44]. However, they require standardized AP radiographs and exact “on film” measurements, either with a ruler or using digital software, which limits their practicability.

Boettner and Zingg reported on a novel technique to determine cup anteversion based on the C-arm tilt angle [45]. This method includes three steps: (1) Taking an AP pelvic view (the obturator foramen need to be symmetric and the sacrum needs to be aligned with the symphysis), (2) centering the C-arm over the hip and gradually tilting away from the operated side until the equatorial plane of the cup lays perpendicular to the receptor and (3) measuring the C-arm tilt angle by direct reading on the C-arm and calculating the anteversion, which is a function of inclination and C-arm tilt angle [45]. Figure 2 shows a final intraoperative C-arm fluoroscopy image after implantation of a THA on the right side.

Compared to CT-scan measurements, the anteversion obtained with this technique was on average 0.2° higher (range −3.0°–3.1°). The average anteversion was 18.4° and the mean inclination was 40.8°; 100% of the cups were implanted within the targeted safe zones [46].

#### 2.2.2. iPad App

According to this method, an iPad software app has been developed [47]. In a five-step procedure, intraoperative acetabular anteversion, inclination and leg length discrepancy are determined based on screenshots of fluoroscopy images [47]. First, the cup is preliminary fixed, and the image is adjusted to meet the criteria defined above, a screenshot is taken and the app provides an inclination reference line that allows for adjustments. The procedure follows the conventional protocol, a second AP-pelvis fluoroscopy image is then taken to determine inclination referenced to the inter-teardrop line and further parameters are assessed after implantation of the femoral component. The mean difference between the intraoperative results and postoperative radiographs were 1.0° for anteversion and 1.2° for inclination, both not being significant [47].

#### 2.2.3. Grids

Alternatively, a grid can be placed on the C-arm to adjust for parallax and measure the corrected component positioning directly (HipGrid™ Drone, OrthoGrid^®^ Systems Inc., Salt Lake City, UT, USA). This increased accuracy of component positioning significantly compared to the sole use of fluoroscopy [48]. Using a digital version of the grid (PhantomMSK™, OrthoGrid^®^ Systems Inc., Salt Lake City, UT, USA) also showed consistent component placement within the defined target zones [49].

#### 2.2.4. Radiographic Overlay Technique

Radiographic overlay technique works with software that processes intraoperative images by comparing the operative side to the contralateral side based on user-set reference points (VELYS™ (formerly JointPoint), DePuy Synthes, West Chester, PEN, USA; Surgeon’s Checklist^®^Hip, Radlink Inc., El Segundo, CA, USA). With each fluoroscopy image, the overlay system generates an ellipse based on the pre-defined targets for the cup position. This can be used as guidance for component placement during the surgery [50,51]. Utilizing such software improved the accuracy of cup positioning compared to cups placed without additional software [52]. Outliers, especially errors in acetabular cup placement, leg length and femoral offset, have been reduced significantly [53].

### 2.3. Mechanical Alignment Guides

The traditional mechanical alignment guide consists of a rod that is applied to the impactor in a way that the desired inclination is achieved if the rod is held horizontal. Alternatively, a spirit level instead of a simple rod can be used. These methods showed superior results to freehand positioning [54].

### 2.4. Patient-Specific Instrumentation

A patient-specific model can be built from preoperative CT or MRI scans. This model is then used to create patient-specific instruments to guide arthroplasty implantation. The guides either work with only mechanical guidance (Signature^TM^, Zimmer Biomet, Warsaw, IN, USA; MyHip^®^, Medacta, Castel San Pietro, CH) or using two laser points (OPS^TM^, Corin Group, Tampa, FL, USA). Positioning accuracy has been reported to be favorable compared to standard instrumentation [55].

### 2.5. Navigation Systems without Imaging

Imageless navigation systems calculate the implant position in relation to a reference plane. This reference usually is the anterior pelvic plane, which needs to be defined by the surgeon who registers required anatomic landmarks at the beginning of the procedure. This step is crucial as inaccuracies at this point will impair guidance and implant positioning.

#### 2.5.1. Optical Alignment

The navigation can be based on optical alignment (IntelliJoint^®^, Kitchener ON, CAN). This system consists of a camera, a probe and a tracker within the sterile field and a workstation with monitor outside of the sterile field. In the beginning of the surgery, a tracker antenna is placed in the anterior superior iliac crest and the surgeon maps further anatomic landmarks to identify the anterior pelvic plane. During acetabular reaming, an additional tracker can be positioned onto the impactor. The workstation computes the relationship between the trackers and the camera and inclination and anteversion are shown on the monitor in real time [56]. The system has shown to be accurate and precise [57]. Comparing intraoperative measurements to postoperative radiographs, 93% of the cups anteversion and inclination were within 10° of the intraoperative measurement [58].

#### 2.5.2. Accelerometer Based Alignment

This system is based on accelerometers and gyroscopes included in both navigation unit and reference sensors which communicate with one another [59]. The registration process with this technique is similar to optical alignment systems. Sensors are fixed to the anterior superior iliac crest; a marker is inserted to the proximal femur and anatomic landmarks are registered using an attached probe. The system can be removed during bony preparation (HipAlign^®^, OrthAlign^®^ Inc., Aliso Viejo CA, USA). After preliminary positioning of the cup, the system is put back in place and zeroed to the pelvis. Once it has calibrated, it is moved on to the acetabular insertion handle. An attached small monitor within the sterile field now shows the acetabular position with both inclination and anteversion on the screen and the component can be adjusted before final implantation.

#### 2.5.3. Inertial Navigation Systems

Inertial navigation systems are a new method of imageless navigations, also based on accelerometers and gyroscopes (Navbit Sprint^®^, Navbit, Sydney, AUS). The system does not use registration of anatomic landmarks for finding the reference plane but uses the patient’s position on the table. Once the patient is positioned on the leveled table, the device is attached to the pelvis by cortical pins. The positioning of the device is not limited to a certain landmark. The operating table then needs to be tilted in a certain sequence; thereby, the system defines the reference axes that serve as a coordinate system. The literature on this device is limited to a few cadaveric studies [60,61].

### 2.6. CT-Based Navigation

Preoperative planning of the component position is performed using CT-scans of the patient’s pelvis. Intraoperatively, an optoelectronic marker/dynamic reference base is attached to the pelvic bone. A standardized protocol with registration of landmarks by the surgeon is then conducted to match the CT-scan based model with the real pelvis. The dynamic reference base is then able to register the pelvis position during the procedure. Both reamer and impactor can be attached with an optoelectronic dynamic reference base as well. A software computes the position of the cup in relation to the pelvis and gives real-time feedback to the surgeon. [62]

### 2.7. Robotic Total Hip Arthroplasty

Robotic THA implantation is based on calibrated preoperative CT scans. A three-dimensional model is created from the images and is intraoperatively used to identify the position of the pelvis based on surface mapping. The first robotic system used in THA was the ROBODOC^®^ (THINK Surgical, Fremont, CA, USA). This system had a fully automated robotic arm that did the femoral-side preparation without the surgeon’s intervention [63]. However, the system was not used for acetabular component positioning. The currently most common technique is robotic arm assisted surgery with a semi-active robotic system (Mako^®^ THA system, Stryker, Kalamzoo, MI, USA). An antenna tracker determines the current position, a robotic arm is positioned in relation to the pelvis during reaming and impaction of the component. The system controls and adjusts to the surgeon’s hand in real time. If the surgeon deviates too far from the preoperative planning, the robotic arm provides resistance and alerts [63]. Robotic-assisted THA showed improvement in accuracy of component positioning, but is associated with additional radiation exposure for the preoperative CT imaging [64,65,66].

## 3. Discussion

Despite the great success of THA in general, misplaced implants can cause severe complications, such as impingement, increased wear, and dislocations [3,4,5,6]. Instability and dislocation have been reported as the main cause for THA revision surgery in the United States, accounting for >20% of all revisions [67].

From the 1970s onward, there has been a continuous discussion about safe zones for cup positioning that may prevent mechanical complications, especially dislocation. The zones initially proposed by Lewinnek et al. were 30–50° inclination and 5–25° anteversion [7]. In 2011, Callanan et al. suggested to target an inclination of 30–45° [68]. Weber et al. suggested even smaller zones, ideally within 40–50° inclination and 20–30° anteversion [14]. The recent development tends towards concepts of patient-specific combined target zones, including the femoral positioning, the patient’s individual anatomy, and the spinopelvic alignment with pelvic tilt and spinal deformities [16,17].

Regardless of what target values are ultimately set in each individual case, the question arises as to how these can best be implemented intraoperatively. The goal is to reproduce preoperative templets and reduce outliers of the different defined safe zone positions. However, this can be challenging. Callanan et al. retrospectively evaluated >2000 THA and found only 50% of the cups implanted within the target zones for both anteversion and inclination [68].

All techniques and methods described above have their advantages and disadvantages and are discussed controversially.

### 3.1. Freehand Positioning and Mechanical Alignment Guides

Freehand positioning has the advantage that anatomic landmarks are patient-specific and mainly independent of the patients’ positioning. However, it has been shown that sagittal pelvic rotation may alter implantation based on the transverse acetabular ligament [34]. Furthermore, correctly locating bony landmarks can be difficult, especially in cases with dysplasia, large osteophytes, or previous trauma [32]. Another factor is the surgeons’ experience level. Suhardi et al. found that deviation of both cup inclination and version was lower for attending surgeons compared to fellows and residents when using freehand positioning. Residents also tended to position cups more horizontal and anteverted compared to fellows [69]. Reported accuracy of freehand positioning varies widely, studies reported implant placement within the targeted safe zone in up to 90% of the cases, others, however, found up to 64% outliers with reference to the Callanan safe zone for cups implanted via a direct anterior approach [29,70]. The accuracy of freehand positioning seems to be inferior compared to implantation with additional navigation [70,71,72].

Additional guidance can be provided by traditional mechanical alignment guides, including inclinometers or patient-specific instruments. Traditional guides are limited to one specific target angle and may lack accuracy. Minoda et al. studied 15 different mechanical alignment guides and found a mean deviation of 6° less anteversion (maximum 12°) and 2° more inclination when comparing the actual to the targeted angles [73]. Using an inclinometer instead of mechanical guides may be more accurate. While some studies found superior results compared to freehand positioning and traditional mechanical guides, especially regarding achieving the targeted inclination and reducing outliers, other studies found no differences [54,71].

Guidance by patient-specific instrumentation may further increase accuracy [55,74]. A recent meta-analysis by Constantinescu et al. reported favorable results for patient-specific instrumentation regarding the deviation from preoperative planning for anteversion, inclination and positioning within the Lewinnek safe zones compared to standard instrumentation [55]. Feretti et al. reported that when using patient-specific instrumentation, 92% of the implants were within 10° deviation from the planned anteversion and inclination, while there was a slight increase in surgical time [74]. Disadvantages are limited choice in implant options, additional costs, and additional radiation exposure due to the planning CT.

In general, however, total hip replacement has been a very successful procedure for the overwhelming majority of patients without the utilization of facilitating technology [2]. In addition, patients with altered spinopelvic alignment or anatomic variations only account for a small proportion of patients undergoing primary total hip replacement and, therefore, the routine use of facilitating technology might not be cost effective for all patients [66,75].

### 3.2. Challenges in Intraoperative Fluoroscopy and How to Address Them

Aiming for better accuracy and intraoperative control, several C-arm based fluoroscopy methods have been developed to determine the cup position intraoperatively. C-arms are easily available in most operating rooms and the costs associated with its use are low. The anterior approach THA with supine patient positioning further facilitates the use of fluoroscopy. The additional radiation exposure is a disadvantage compared to freehand positioning or mechanical guides. However, the mean patient exposure during anterior approach THA is 178 mrem and below a single pelvic radiograph (600 mrem) [76]. The dosage has been reported to be likely negligible [77].

The main challenge when using fluoroscopy are stereometric effects (parallax) affecting projections and measurements. Therefore, the intraoperative C-arm measurements may not resemble the measurements on postoperative standard pelvic radiographs. Deviation of the central X-ray beam from the cup alters the measured positioning [78]. This effect is especially relevant in C-arm imaging as the decreased field of view causes that the cups are projected at the outer boarders of the image. This has shown to reduce intraoperatively measured inclination compared to measurements obtained from postoperative images [38]. Furthermore, measured acetabular version differs between standing and supine pelvic radiographs within in the same patient, mainly due to changes in the pelvic tilt [79]. A pelvic roll back of 1° has shown to cause 0.7° functional anteversion of the cup [80]. Differences in pelvic tilt can be anticipated by measuring the distance between the pubic symphysis and the sacroiliac-joint-line and applying correction factors according to Tannast et al. [39]. Furthermore, the sacrum should be in line with the symphysis and the obturator foramen should project symmetrically in order to avoid error due to rotation [38].

Several methods have been described to obtain anteversion from standardized pelvic radiographs. The reliability of these methods has been confirmed but their feasibility in the intraoperative setting with C-arm imaging is limited as they require standardized AP views and exact direct measurements with either a ruler or a specific software [44,45]. The effects of parallax and patient positioning described above are likely to further impair measurements with these methods when performed intraoperatively on C-arm fluoroscopy images.

To adjust to the specific challenges of intraoperative fluoroscopy, using grids, overlaying radiography or mathematical correction equations and the c-arm tilt angle may be beneficial.

#### 3.2.1. Grids

Gililland et al. aimed to address parallax effects by adding a grid that was fixed to the operation table [81]. Using the grid additionally compared to fluoroscopy alone, they found superior results for meeting the abduction target and restoring offset and leg length. They were further able to reduce the elapsed surgical time from a mean of 94 min before adding the grid technique to 79 min with the grid [81]. However, in order to adequately use the technique, it is necessary to adjust the grid with anatomic references such as the teardrops, this may require moving the patient on the table or leave the sterile field to adjust the grid itself. Newer versions with a grid fixed to the C-arm allow for image adjustment without moving the patient. However, the manual method still requires several images until grid and patient are adjusted in a way that the reference points are projected adequately. This can be avoided with a digital grid version that allows to choose symmetric reference points from only one image. Thorne et al. reported a mean surgical time of 71 and 69 min for these manual and digital versions, respectively [49]. However, the generalizability of these results needs to be further investigated in a setting including less experienced surgeons. A limitation to grid systems is that they do not provide guidance for anteversion.

#### 3.2.2. Radiographic Overlay Technique

Software with digital radiographic overlay technique can address all significant parameters related to implant placement. Thorne et al. compared a grid (PhantomMSK™, OrthoGrid^®^ System Inc., Salt Lake City, UT, USA) to a digital radiographic overlay technique (Radlink Inc., Los Angeles, CA, USA) [50]. The targeted anteversion was more likely to be achieved with the overlay method, differences for abduction, global hip offset, and leg length discrepancy were not significant. After starting to use a digital imaging system, Hambright et al. found a significant reduction in outliers for leg length discrepancy and femoral offset, a trend towards reduced outliers for inclination was not statistically significant [82]. Penenberg et al. reported on patients who underwent posterior approach THA using the Radlink^TM^ system and found 97.8% placed within 30–50° of abduction and the remaining cases placed outside the zone, however purposefully [83]. Only a minor increase in the mean additional operation time of 2–5 min due to the digital radiographs has been reported [82,83]. Though, Thorne et al. found a longer operating time when using the overlay method compared to when using a grid [50].

Both methods, grids and digital overlay technique, require additional devices and/or software.

#### 3.2.3. Fluoroscopy without Additional Applications

Boettner and Rueckl found that intraoperative determination of cup inclination from fluoroscopic images was reliable and reproducible. The problem of measurement deviance from postoperative radiographs could be addressed efficiently by applying a correction factor of 5° [38]. This technique requires a correction for pelvic tilt using the method described by Tannast et al. [39]. Furthermore, the sacrum needs to be projected in line with the symphysis and the obturator foramen should be symmetrical to avoid rotation errors. The number of the correction factor may vary when using different C-arm models [38]. To allow simple intraoperative determination of cup anteversion without additional devices, Boettner and Zingg developed a three-step measurement procedure utilizing the correlation between the C-arm tilt angle and cup anteversion. First, the pelvis was placed in a neutral position and the abduction angle was evaluated, the C-arm was then centered over the hip and tilted away until the ellipse formed by the acetabular rim turned into a line. The C-arm tilt angle could then be read on the device. This allowed for calculation of the cup anteversion using a mathematical equation based on C-arm tilt angle and inclination [46]. When comparing the intraoperative measurements to postoperative CT scans, this method showed highly accurate results as long as a neutral pelvis position was assured [45,46]. The accuracy of cup placement showed superior results in anterior THA compared to posterior THA [46]. According to this protocol, a software app for the iPad has been developed using pictures of the fluoroscopy images. Additionally, the app provides optic guidance with projected reference lines and showed reliable results for accurate measurement of acetabular component position and leg length [47].

### 3.3. Imageless Navigation

Imageless navigation systems provide the advantage of avoiding additional radiation exposure to patient and surgeon. However, they require additional hard- and software and intraosseous placements of pins for trackers. The main challenge, and source of errors, is the identification of anatomic landmarks. As described in freehand positioning procedures, intraoperative identification of bony landmarks can be difficult in cases with uncommon anatomy due to trauma, dysplasia or osteophytes. Percutaneous identification can be altered by increased soft tissue or deviations in patient positioning [32]. Furthermore, increased spinopelvic mobility has been found to be a risk factor for errors in cup positioning, yet this has only been reported when operating the patient in lateral decubitus position [84]. Imageless navigation is associated with longer operation time compared to conventional THA [85]. Brown et al. reported additional time demand especially during the pre-operative phases, specifically to re-drape and re-position the patient after the placement of the initial tracker and to register the landmarks [86].

Imageless navigation systems can reduce fluoroscopy exposure time and showed more cups with inclination within the safe zone and a reduced number of outliers compared to non-navigated THA (OrthoPilot^®^, B. Braun Aesculap, Tuttlingen, GER and Naviswiss Hip miniature imageless navigation platform, Naviswiss AG, Bruss, CHE) [87,88]. Comparing navigation derived intraoperative angles (Stryker Navigation System, Stryker, Kalamazoo, MI, USA) to postoperative CT scans, the cups were placed within a maximum deviation of 10° in 97% for inclination and 77% for abduction [89]. A recent meta-analysis by Zurmühle et al. found that using imageless navigation, cups were generally impacted in higher inclination compared to the target value (+3.2°), the difference for anteversion (+5.0°) was not significant. Including this mean deviation, the values were still within the safe zones [90]. Another meta-analysis published by Migliorini et al. in 2022 found slightly lower leg length discrepancies for navigated THA, but similar results for cup positioning and dislocation rates when comparing conventional and navigated THA [85].

Inaccuracies arising from difficulties with the identification of anatomic landmarks may be addressed by systems that use the operating table tilt to generate a reference system and only require one sensor to be fixed to pelvis without stipulating a certain position for fixation (Navbit Sprint^®^, Navbit, Sydney, Australia). A cadaveric study by Shatrov et al. found a difference of 2.3° for acetabular version when comparing the devices measurements to postoperative CT-scans [61]. Accuracy for cup positioning with the device seems to be good and the registration method using the table tilt has been validated [60,61]. However, the literature on this technique is very limited and it needs to be validated in larger cohorts of living patients.

### 3.4. Robotic-Assisted Hip Replacements

Robotic-assisted arthroplasty surgery has gained popularity over the past years. Whether it leads to increased precision compared to conventional THA is under debate. Several studies found more cups implanted within the Lewinnek safe zones compared to conventional THA [91,92]. Yet, this was not always statistically significant [93]. In a recent randomized controlled trial, Guo et al. found an increased operation time with robotic assistance [93]. However, they reported a learning curve of 13 operations after which the operation time decreased significantly [93]. Learning curves are reported regarding placement accuracy as well [92]. A higher rate of blood transfusions has been found by Remily et al. whereas others found no difference in intraoperative bleeding between robotic-assisted and conventional procedures [93,94]. The device is associated with relevant initial cost. However, the procedure may be overall slightly more cost efficient compared to conventional THA [94,95]. As it requires preoperative CT-imaging, robotic-assisted surgery brings higher radiation exposure compared to imageless, fluoroscopy guided or freehand positioning.

## 4. Conclusions

When choosing the positioning method, various factors, such as operating time, radiation exposure, costs and downsides of additional soft- and hardware, and the limited choice of implants with certain techniques, need to be weighed against each other. Aiming for precise cup positioning in anterior THA, additional guidance seems to generally provide superior results compared to sole freehand positioning. Younger and low-volume surgeons tend to have more difficulties with precise positioning then experienced high-volume surgeons and may especially benefit from more intraoperative guidance and feedback during the procedure [69,96]. A potential pitfall in most techniques is the correct identification of anatomic landmarks; therefore, this step needs to be conducted with special caution and precision. Intraoperative fluoroscopy techniques are less prone to these errors; however, it is crucial to adjust the image correctly to avoid inaccuracies due to pelvic tilt, rotation and parallax. The senior author regularly uses intraoperative C-arm imaging, calculating the anteversion via the C-arm tilt angle and adjusting inclination with a correction factor of 5°, which has been proven to be a reliable method [38,46].

## 5. Future Directions

The rapid development in software engineering may soon simplify the application of, e.g., mathematical corrections and allow such methods to be implemented more precisely and quickly. The need for various additional devices may be reduced by the increased use of regularly available devices, such as tablets. This offers the chance to facilitate a wider use outside of highly specialized centers. This seems favorable as especially low-volume surgeons may benefit from more guidance in cup positioning. New registration methods avoiding the necessity of exactly locating anatomic landmarks may reduce this source of inaccuracy in the future. However, at this point, data on the validity of these devices is very limited. Augmented reality may also start playing a role in navigation during surgery. The first of such prototypes have already been validated in cadaveric studies [97].

## Figures and Tables

**Figure 1 medicina-59-00271-f001:**
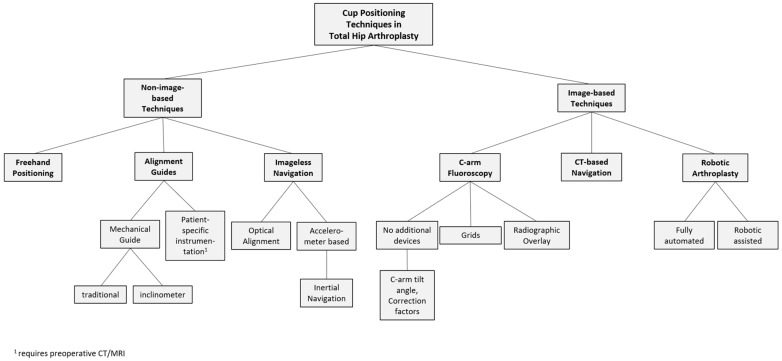
Organizational diagram showing an overview of various cup positioning techniques in total hip arthroplasty. Patient-specific instrumentation is listed with alignment guides, however, it requires preoperative imaging to manufacture the instrumentation. See the main text for detailed information on the techniques.

**Figure 2 medicina-59-00271-f002:**
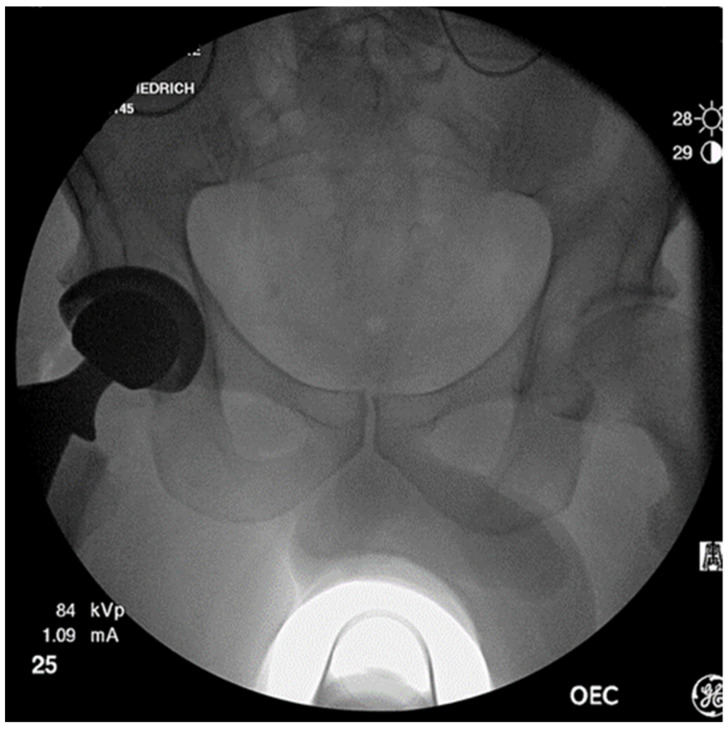
Intraoperative C-arm fluoroscopy image of a pelvis in antero-posterior projection after implantation of a total hip arthroplasty on the right side. The sacrum is in line with the pelvic symphysis and the obturator foramen are projected equally to avoid errors due to rotation.

## Data Availability

Not applicable.

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
