# Peer review of "Achieving Precise Cup Positioning in Direct Anterior Total Hip Arthroplasty: A Narrative Review"

_medicina, 2023, doi:10.3390/medicina59020271_

Round 1

Reviewer 1 Report

The article "Achieving precise cup positioning in direct anterior total hip arthroplasty" is devoted to the study of existing navigation systems and methods for improving the accuracy of cup positioning in hip arthroplasty in order to help both experienced and inexperienced surgeons perform this operation. Various techniques have been described based on freehand positioning by anatomical landmarks, C-arm fluoroscopy, imageless navigation, or robotic-assisted-surgery. This article aims to provide an overview of the various methods and systems, with their specific benefits and potential pitfalls, as well as a glimpse into the future of cup positioning in anterior hip arthroplasty. And even in the list of references there are more than 50% of the sources that have been published over the past 5 years. And, based on the above list of references, an extensive discussion of the corresponding navigation systems was carried out.

But it is necessary to point out the following serious shortcoming of the review.

The review cannot be considered reliable, since it is made in a non-standard free form and it is not indicated in which databases, for example, Scopus, WoS, PubMed, EBSCO, the search was carried out, what keywords were used, what filters were used for selection, whether search on patents for inventions to talk about advanced navigation systems, etc.

But it is known that for such a review of publications in the area of medicine, a special Prisma standard (doi: 10.1186/s13643-020-01542-z) was created, the use of which would make it possible to give a good reliable review and achieve the indicated goal in the article. In almost all Review articles on medical topics, the authors use this selection system.

Therefore, the article requires serious revision and, in case of its absence, is not recommended for publication.

In addition, it is recommended to read the text of the article carefully, as there are the following technical notes.

1. On line 109, the word at the beginning of a sentence begins with a lowercase letter.

2. In the Conclusion section, the meaning of the phrase in lines 440-441 is not clear, since after the word parallax there is a dot, and then the word begins with a lowercase letter.

3. It is unclear what the same format should be in the title text in lines 314, 335, 353, 376, 413.

4. You need to check lines 384-385.

Author Response

Reviewer’s remark

Authors‘ response

Changes to the manuscript

Reviewer 1

The article "Achieving precise cup positioning in direct anterior total hip arthroplasty" is devoted to the study of existing navigation systems and methods for improving the accuracy of cup positioning in hip arthroplasty in order to help both experienced and inexperienced surgeons perform this operation. Various techniques have been described based on freehand positioning by anatomical landmarks, C-arm fluoroscopy, imageless navigation, or robotic-assisted-surgery. This article aims to provide an overview of the various methods and systems, with their specific benefits and potential pitfalls, as well as a glimpse into the future of cup positioning in anterior hip arthroplasty. And even in the list of references there are more than 50% of the sources that have been published over the past 5 years. And, based on the above list of references, an extensive discussion of the corresponding navigation systems was carried out. But it is necessary to point out the following serious shortcoming of the review.

The review cannot be considered reliable, since it is made in a non-standard free form and it is not indicated in which databases, for example, Scopus, WoS, PubMed, EBSCO, the search was carried out, what keywords were used, what filters were used for selection, whether search on patents for inventions to talk about advanced navigation systems, etc.

But it is known that for such a review of publications in the area of medicine, a special Prisma standard (doi: 10.1186/s13643-020-01542-z) was created, the use of which would make it possible to give a good reliable review and achieve the indicated goal in the article. In almost all Review articles on medical topics, the authors use this selection system.

Therefore, the article requires serious revision and, in case of its absence, is not recommended for publication.

Thank you for your thorough review.

Medicina distinguishes between review and systematic review in its article categories (https://www.mdpi.com/about/article_types). While a systematic review is suggested to follow the PRISMA checklist, this is not required for “simple” reviews.

We changed the manuscript by adding “a narrative review” to the title in order to

emphasize the character of this review more clearly.

Examples for recently published medical articles of the same category are:

Wu, X.; Zhao, L.; Li, K.; Yang, J. The Role of NLRP3 Inflammasome in IgA Nephropathy. Medicina 202359, 82. https://doi.org/10.3390/medicina59010082

Radu, P.; Zurzu, M.; Paic, V.; Bratucu, M.; Garofil, D.; Tigora, A.; Georgescu, V.; Prunoiu, V.; Popa, F.; Surlin, V.; Strambu, V. Interstitial Cells of Cajal—Origin, Distribution and Relationship with Gastrointestinal Tumors. Medicina 202359, 63. https://doi.org/10.3390/medicina59010063

Serrano, M.A.; Gomes, A.M.C.; Fernandes, S.M. Monitoring of the Forgotten Immune System during Critical Illness—A Narrative Review. Medicina 202359, 61. https://doi.org/10.3390/medicina59010061

Ogilvie, A.; Kim, W.J.; Asirvatham, R.D.; Fontalis, A.; Putzeys, P.; Haddad, F.S. Robotic-Arm-Assisted Total Hip Arthroplasty: A Review of the Workflow, Outcomes and Its Role in Addressing the Challenge of Spinopelvic Imbalance. Medicina 202258, 1616. https://doi.org/10.3390/medicina58111616

Maffulli, N.; Aicale, R. Proximal Femoral Fractures in the Elderly: A Few Things to Know, and Some to Forget. Medicina 202258, 1314. https://doi.org/10.3390/medicina58101314

Beckers, G.; Djebara, A.-E.; Gauthier, M.; Lubbeke, A.; Gamulin, A.; Zingg, M.; Bastian, J.D.; Hannouche, D. Acetabular Peri-Prosthetic Fractures—A Narrative Review. Medicina 202258, 630. https://doi.org/10.3390/medicina58050630

Achieving precise cup positioning in direct anterior total hip arthroplasty: A narrative review.

On line 109, the word at the beginning of a sentence begins with a lowercase letter.

The letter has been revised.

Measuring the C-arm tilt angle by direct reading on the C-arm and calculating the anteversion, which is a function of inclination and C-arm tilt angle.

In the Conclusion section, the meaning of the phrase in lines 440-441 is not clear, since after the word parallax there is a dot, and then the word begins with a lowercase letter.

The half sentence following the word parallax was mistakenly kept from the medicina word template. It has been removed from the manuscript.

Intraoperative fluoroscopy techniques are less prone to these errors, however it is crucial to adjust the image correctly to avoid inaccuracies due to pelvic tilt, rotation and parallax.

It is unclear what the same format should be in the title text in lines 314, 335, 353, 376, 413.

“Grids”, “Radiographic overlay technique”, and “Fluoroscopy without additional applications” are subheadings of “Challenges in intraoperative fluoroscopy and how to address them” and therefore formatted in template heading no 3.

“Imageless navigation” and “Robotic-assisted hip replacements” have been changed to format heading 2.

Lines 391 and 429

You need to check lines 384-385.

The additional blank line has been removed.

Furthermore, increased spinopelvic mobility has been found to be a risk factor for errors in cup positioning, yet this has only been reported for operating the patient in lateral decubitus position

Reviewer 2 Report

  1. What is the main question addressed by the research?

The specific question of the research is correct acetabular cup position in the total hip arthroplasty.

  1. Do you consider the topic original or relevant in the field? Does it

address a specific gap in the field?

The topic is discussed as a review and addresses the specific gaps in each of the existing methods to achieve the goal of proper cup position.

  1. What does it add to the subject area compared with other published

material?

The manuscript covers all existing methods in a well-structured and organised manner. All important references are included and addressed thoroughly.

  1. What specific improvements should the authors consider regarding the

methodology? What further controls should be considered?

the manuscript is well written and requires no specific improvements. No controls should be considered.

  1. Are the conclusions consistent with the evidence and arguments presented

and do they address the main question posed?

The conclusions are consistent with evidence and arguments presented and do address the main question posed.

  1. Are the references appropriate?

References are more than appropriate.

  1. Please include any additional comments on the tables and figures.

Introductory figure is appropriate.

     I commend the authors for performing their research entitled "Achieving precise cup positioning in direct anterior total hip arthroplasty". I found the review very interesting for practicing orthopedic surgeons, comprehensive and well written. All available systems for achieving desired acetabular cup position are mentioned. Discussion is critical and points out the pros, the cons and limitations of every method explained in the previous sections. Conclusions are sound with respect to the future developments. The authors' personal choice of one method that suited them best would be much appreciated.

Author Response

Reviewer’s remark

Authors‘ response

Changes to the manuscript

Reviewer 2

 I commend the authors for performing their research entitled "Achieving precise cup positioning in direct anterior total hip arthroplasty". I found the review very interesting for practicing orthopedic surgeons, comprehensive and well written. All available systems for achieving desired acetabular cup position are mentioned. Discussion is critical and points out the pros, the cons and limitations of every method explained in the previous sections. Conclusions are sound with respect to the future developments. The authors' personal choice of one method that suited them best would be much appreciated.

Thank you for your thorough review.

The senior author’s recommended technique has been added to the manuscript at the end of the conclusion.

The senior author regularly uses intraoperative c-arm imaging, calculating the anteversion via the c-arm tilt angle and adjusting inclination with a correction factor of 5°, which has been proven to be a reliable method.

Reviewer 3 Report

Dear authors, I would like to congatualte for an excellent overview over this very relevant topic, that will create a lot of citations.

I have only a few suggestions of publications that are relevant to the topic you may consider to incorporate: 

PMID: 33687529

PMID: 33582865

PMID: 35776176

PMID: 35652949

PMID: 34643783

Author Response

Reviewer’s remark

Authors‘ response

Changes to the manuscript

Reviewer 3

Dear authors, I would like to congatualte for an excellent overview over this very relevant topic, that will create a lot of citations.

I have only a few suggestions of publications that are relevant to the topic you may consider to incorporate: 

PMID: 33687529

PMID: 33582865

PMID: 35776176

PMID: 35652949

PMID: 34643783

Thank you for your thorough review and the additional literature suggestions.

Jungwirth-Weinberger et al. (PMID 33687529) has been added as a reference. à reference number 25.

Bechler et al. (PMID 33582865) has been referenced in section 2.2.2 iPad app.

Ong et al. (PMID 35776176) has been added as a reference. à reference number 88.

Kunze et al. (PMID 35652949) reported on discharge and thrombogenic markers in patients undergoing unilateral total hip arthroplasty. The study provides interesting findings on the comparison of anterior posterior approach THA for these aspects, however since the study did not fully match the topic of the current review, we decided against including this reference.

Dimitriou et al. (PMID 34643783) found that dislocation rates following anterior THA are not increased in patients with spinopelvic fusion compared to patients without spinopelvic fusion. However it cannot be definitely said from the methods section what cup positioning technique has been used in the study. We therefore decided against including this reference.

To which extent the surgical approach effects the accuracy of cup positioning is under discussion. Some studies showed superior results for THA implanted via a direct anterior approach compared to posterior or lateral approaches and also reported lower dislocation rates.

Imageless navigation systems can reduce fluoroscopy exposure time and showed more cups with inclination within the safe zone and a reduced number of outliers compared to non-navigated THA (OrthoPilot®, B. Braun Aesculap, Tuttlingen, GER and Naviswiss Hip miniature imageless navigation platform, Naviswiss AG, Bruss, CHE).

Round 2

Reviewer 1 Report

I am satisfied with the authors' responses, in particular due to the change in the manuscript title, and endorse the publication in present form.